 # The Art of SOCRATIC QUESTIONING: Recursive Thinking with Large Language Models

**Jingyuan Qi**[*]♠    **Zhiyang Xu**[*]♠    **Ying Shen**[†]♠    **Minqian Liu**[†]♠
**Di Jin**♡    **Qifan Wang**♣    **Lifu Huang**♠

♠Virginia Tech    ♡Amazon Inc.    ♣Meta AI
{jingyq1,zhiyangx,yings,minqianliu,lifuh}@vt.edu
djinamzn@amazon.com    wqfcr@meta.com

## Abstract

Chain-of-Thought (CoT) prompting enables large language models to solve complex reasoning problems by generating intermediate steps. However, confined by its inherent single-pass and sequential generation process, CoT heavily relies on the initial decisions, causing errors in early steps to accumulate and impact the final answers. In contrast, humans adopt recursive thinking when tackling complex reasoning problems, i.e., iteratively breaking the original problem into approachable sub-problems and aggregating their answers to resolve the original one. Inspired by the human cognitive process, we propose SOCRATIC QUESTIONING, a divide-and-conquer style algorithm that mimics the recursive thinking process. Specifically, SOCRATIC QUESTIONING leverages large language models to raise and answer sub-questions until collecting enough information to tackle the original question. Unlike CoT, SOCRATIC QUESTIONING explicitly navigates the thinking space, stimulates effective recursive thinking, and is more robust towards errors in the thinking process. Extensive experiments on several complex reasoning tasks, including MMLU, MATH, LogiQA, and visual question-answering demonstrate significant performance improvements over the state-of-the-art prompting methods, such as CoT, and Tree-of-Thought. The qualitative analysis clearly shows that the intermediate reasoning steps elicited by SOCRATIC QUESTIONING are similar to humans' recursively thinking process of complex reasoning problems[12].

## 1 Introduction

*The art of Socratic Questioning is important for critical thinkers and excellence of thought. What Socratic adds is systematicity, depth, and a keen*

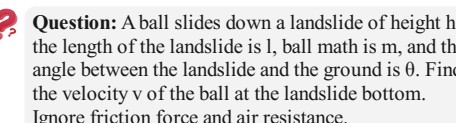

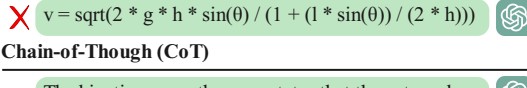

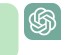

Figure 1: Example of a complex question solved by the Standard Prompting, Chain-of-Thought, and SOCRATIC QUESTIONING. Accumulated incorrect reasoning are highlighted in red.

*interest in assessing the plausibility of things.*

- L. ELDER and R. PAUL, 1998

One unique capability that allows humans to excel at solving complex reasoning problems is *recursive thinking*. If the answer is not immediately achievable, humans think deeper by recursively decomposing the complex problem into simpler and solvable sub-problems.

Recently, by scaling up the parameters, large-

---

[1][*] Co-first Authors, † Co-second Authors

[2]All the programs and necessary resources are released in https://github.com/VT-NLP/SOCRATIC-QUESTIONING

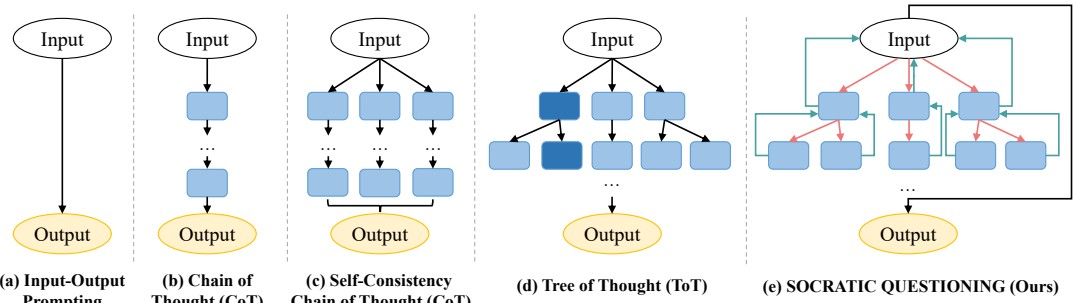

Figure 2: **Schematic comparison of various prompting methods.** Each blue rectangle box represents a *thought* serving as an intermediate reasoning step in the problem-solving process. SOCRATIC QUESTIONING incorporates both a top-down exploration process (in red line) to deconstruct complex problems into smaller sub-questions and a bottom-up backtracking process (in green line) to recursively solve these sub-questions and gather solutions for higher-level problems.

scale language models (LLMs) (Brown et al., 2020; Chung et al., 2022; OpenAI, 2022; Touvron et al., 2023) gain emerging capabilities, such as Chain-of-Thought (CoT) (Wei et al., 2022) which decomposes the complex problem and solves it step by step. Though CoT has been proven to be effective on various complex reasoning tasks, it's in nature a single-pass and sequential thinking process that generates the next step based on previous steps, thus only exploring a single way of thinking to approach a problem and easily accumulating errors from previous steps (Turpin et al., 2023). In addition, CoT lacks the ability to refine the already generated reasoning path, as shown in Figure 1.

Inspired by the recursive thinking of humans, we propose SOCRATIC QUESTIONING, a novel divide-and-conquer fashion algorithm that prompts language models to solve complex reasoning problems. As shown in Figure 2 (e), SOCRATIC QUESTIONING consists of a top-down exploration process and a bottom-up backtracking process. Specifically, in the top-down exploration process, the original complex problem is decomposed into simpler or related sub-problems until the sub-problems can be solved. In the bottom-up backtracking process, the solutions to the sub-problems are returned and selectively used to solve the original problem. The fundamental component that drives SOCRATIC QUESTIONING is a SELF-QUESTIONING (SQ) module, that leverages large-scale language models to proactively raise and answer questions that are essential to solving the target question. SOCRATIC QUESTIONING recursively backtracks and tailors the intermediate thoughts acquired from SELF-QUESTIONING until reaching an answer to the original input question. It explicitly navigates the thinking space and is more robust towards thinking errors compared with pre-

vious prompting methods including CoT, Self-Consistency Chain-of-Thought (Wang et al., 2023), and Tree-of-Thought (Yao et al., 2023), as shown in Figure 2.

To show the effectiveness of SOCRATIC QUESTIONING, we conduct extensive experiments on various complex reasoning tasks including the chemistry and physics tasks (Hendrycks et al., 2020), mathematical tasks (Hendrycks et al., 2021), and reading comprehension tasks (Liu et al., 2020). Additionally, we showcase the generalizability of our method by conducting experiments with few-shot multimodal reasoning on VQA-V2 (Goyal et al., 2017), OK-VQA (Marino et al., 2019), and AOK-VQA (Schwenk et al., 2022) datasets. Experimental results indicate that SOCRATIC QUESTIONING substantially improves performance over CoT, SC-CoT, and ToT across all language tasks and outperforms several strong baselines in few-shot multimodal reasoning. The qualitative analysis further demonstrates that SOCRATIC QUESTIONING is capable of eliciting the intermediate reasoning steps through SELF-QUESTIONING, like a critical thinker, and solving complex reasoning problems. The main contributions of our paper are as follows:

- We propose SOCRATIC QUESTIONING, a novel prompting algorithm that can navigate the cognitive thinking space in a ***recursive*** manner.
- We introduce the SELF-QUESTIONING module, a core component that actively probes complex problems from various perspectives by raising and addressing questions essential for solving the main problem.
- Our approach achieves significant improvements over the previous prompting methods in various complex reasoning tasks.

## 2 Related Work

**Prompting Large Language Models** With the scaling of both modal size and corpus size, large language models (LLMs) such as GPT-3 (Brown et al., 2020) and ChatGPT (OpenAI, 2022) have exhibited emergent abilities, including prompting (Brown et al., 2020), in-context learning (Dong et al., 2023), and commonsense reasoning (Wei et al.). One notable example of emergent abilities is the Chain-of-Thought (CoT) (Wei et al., 2022) which steers large language models to resolve complex problems by guiding them to produce a sequence of intermediate steps before giving the final answer. Self-Consistency Chain-of-Thought (SC-CoT) (Wang et al., 2023) improves naive CoT by sampling multiple reasoning paths and selecting the most consistent answer. SC-CoT is based on the assumption that given a complex reasoning problem, multiple reasoning paths can lead to the unique correct answer. Tree-of-Thought (ToT) (Yao et al., 2023) proposes to break the thinking process into small steps and at each step, the language model deliberately decides a set of next steps to try.

**Multimodal Reasoning with Large Language Models** Recent studies have explored the collaboration among diverse language and visual models (Yang et al., 2022; Zeng et al., 2022; Huang et al., 2022). For example, PICa (Yang et al., 2022) utilize image captions as the bridge between visual model and GPT-3 to peform few-shot knowledge-based VQA. Socratic models (Zeng et al., 2022) present a modular framework that utilizes language-based exchange between pre-trained models and other modules. However, these studies only rely on text as the shared interface, which can inevitably lead to information loss when translating visual information into language. In addition, several concurrent studies (Wu et al., 2023; Surís et al., 2023; Lu et al., 2023) have also explored the utilization of large language models for composing various language and visual models.

**Question Decomposition** Recent research has underscored the effectiveness of question decomposition and sub-question generation techniques in tackling complex tasks. DECOMPRC (Min et al., 2019), for instance, utilizes a limited amount of human-labeled data to train a span-based sub-question generator and simplifies multi-hop questions into single-hop questions. Similarly, (Nogueira and Cho, 2017) leverages reinforcement learning for weakly supervised question generation and (Perez et al., 2020) introduces ONUS, an algorithm that harnesses large-scale questions sourced from the internet to perform unsupervised question decomposition. More recently, (Patel et al., 2022) proposes an alternative approach to enhance the performance of LLMs by decomposing challenging questions into simpler sub-questions on various tasks. Notably, the efficacy of question decomposition has been demonstrated across a range of tasks and domains, including solving mathematical problems (Shridhar et al., 2022), medical question answering (Roberts et al., 2014), and factual correction (Huang et al., 2023).

## 3 Method

### 3.1 SOCRATIC QUESTIONING

Figure 3 shows the overview of the SOCRATIC QUESTIONING approach, which is essentially a recursive thinking process involving a ***top-down exploration process*** (in red line) and a ***bottom-up backtracking process*** (in green line). The top-down exploration process proactively breaks down the question into simpler sub-questions until the sub-questions are answered with high confidence. The bottom-up backtracking process recursively solves questions in which the answers to sub-questions are collected to solve the higher-level more complex questions.

In the beginning, we are given a target question $Q_1^{0,0}$, the context $C$ (if provided), and an optional hint $H_1^{0,0}$. The hint is initially Null but will be updated and enriched as the recursive thinking process continues and results from sub-questions are aggregated. We first run the top-down process to explore the thinking space by invoking the SELF-QUESTIONING module. We use depth $d$ and turn $t$ to identify the node in our reasoning tree. Depth $d$ refers to the traditional depth of the recursion algorithm. Turn $t$ refers to the times of SOCRATIC QUESTIONING invoking the SELF-QUESTIONING module for each question. For example, at depth $d$, turn $t$, SELF-QUESTIONING takes in the $i_{th}$ question $Q_i^{d,t}$, hint $H_i^{d,t}$, the context $C$, and decides if it can answer the question $Q_i^{d,t}$: (1) If SELF-QUESTIONING can directly output the answer $A_i^{d,t}$ for the question $Q_i^{d,t}$ with high confidence, the bottom-up backtracking process starts by converting the answer $A_i^{d,t}$ to a hint $\widetilde{H}_i^{d,t}$ with a QA-to-Hint module ($\widetilde{H}_i^{0,t}$ equals $A_i^{0,t}$ directly when $d =$

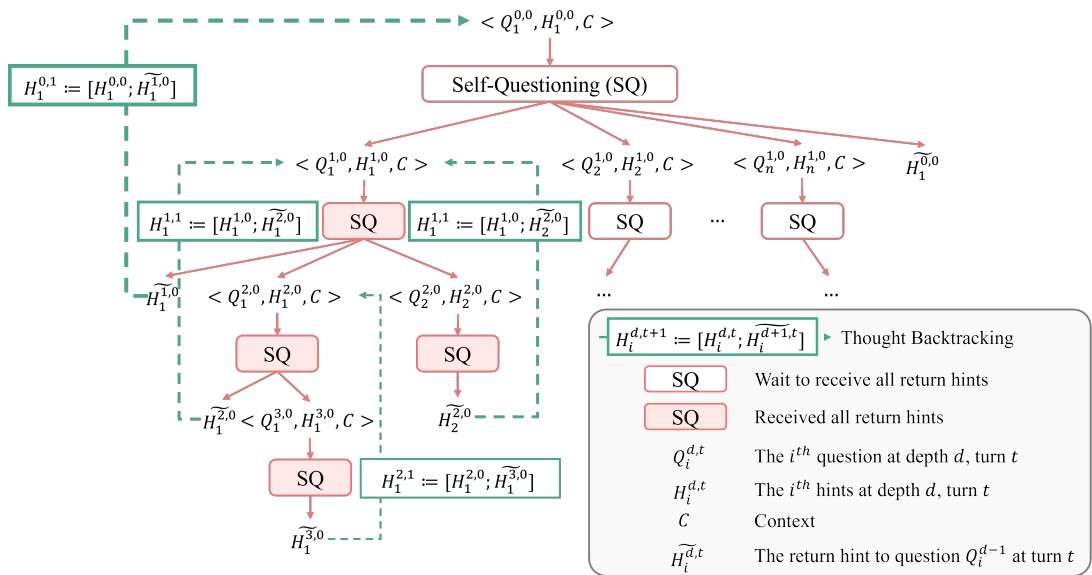

Figure 3: Overview of our SOCRATIC QUESTIONING algorithm.

0) and adding $\widetilde{H}_i^{d,t}$ into the hints $H^{d-1,t}$ of the parent question $Q^{d-1}$. (2) If SELF-QUESTIONING cannot directly output an answer with high confidence, it outputs a set of sub-questions $\mathcal{Q}^{d+1,t}$ related to $Q_i^{d,t}$. Then we run SELF-QUESTIONING on each newly generated sub-question $Q_j^{d+1,t}$ until it's answered with high confidence. Once we obtain the answers to all the sub-questions $\mathcal{Q}^{d+1,t}$, we convert the answers into hints and incorporate them to update $H_i^{d,t}$ to $H_i^{d,t+1}$. We then run SELF-QUESTIONING on $Q_i^{d,t+1}$ again with updated hints $H_i^{d,t+1}$. This recursive process continues until we reach the tree's root and the original question $Q_1^0$ is answered by $\tilde{H}_1^0$. We provide the pseudo-code of SOCRATIC QUESTIONING in Algorithm 1.

## 3.2 SELF-QUESTIONING

SELF-QUESTIONING is designed to answer the given question, self-check the answer, and raise sub-questions. At depth $d$, turn $t$, SELF-QUESTIONING takes in the $i_{th}$ question $Q_i^{d,t}$, the context $C$ (if available), and hints $H_i^{d,t}$ (if available) and tries to generate an answer or a set of related sub-questions. SELF-QUESTIONING consists of two modules, a **Question-Answering (QA) Module** that outputs an answer $A_i^{d,t}$ for $Q_i^{d,t}$ based on $C$ and $H_i^{d,t}$, and an associated confidence level: high, medium, or low. If the confidence of the answer is high, or either depth $d$ or turn $t$ met the pre-defined limit $d_m$ and $t_m$, SELF-QUESTIONING invokes the QA2H module to merge the question $Q_i^{d,t}$ and answer $A_i^{d,t}$ to hint $\widetilde{H}_i^{d,t}$ as output (when $d = 0$, we skip the merging process because the

---

**Algorithm 1:** SOCRATIC QUESTIONING

**Input:** Question $Q_i^{d,t}$, Hint $H_i^{d,t}$, Context $C$, Current Depth $d$, Max Depth $d_m$, Current Turn $t$, Max Turn $t_m$, Question Answer Prompt $P_{QA}$, Question Generate Prompt $P_{QG}$, QA to Hint Prompt $P_{QA2H}$

**Output:** Hint $\widetilde{H}_i^{d,t}$

1  **for** $t \leq t_m$ **do**
2    // call self-questioning
   $< \mathcal{Q}^{d+1,t}, \mathcal{H}^{d+1,t}, C > \leftarrow$
   SELF-QUESTIONING$(Q_i^{d,t}, H_i^{d,t}, C, d, d_m, t, t_m, P_{QA}, P_{QG})$ ;
3    **if** $\mathcal{Q}^{d+1,t} \neq \emptyset$ **then**
4      **for** each $Q_j^{d+1,t} \in \mathcal{Q}^{d+1,t}$ **do**
      // recursively answer sub-questions
5       $\widetilde{H}_j^{d+1,t} \leftarrow$
      SOCRATIC QUESTIONING$(Q_j^{d+1}, H_j^{d+1,t}, C, d+1, d_m, t, t_m, P_{QA}, P_{QG})$;
      // gather hint
6       $H^d$.insert($\widetilde{H}_j^{d+1,t}$));
7    **else**
8      $\widetilde{H}_j^{d,t} \leftarrow \mathcal{H}^{d+1,t}[0]$;
9      **return** $\widetilde{H}_j^{d,t}$;
10   $t \leftarrow t + 1$;

---

answer $A_1^{0,}$ is the final answer and does not need to be rewritten to hint). Both Max Depth $d_m$ and Max Turn $t_m$ prevent SOCRATIC QUESTIONING from infinite recursion. On the other hand, if the confidence of the answer is lower than high, a **Question-Generation (QG) Module** is called to generate a set of sub-questions $\{Q_0^{d+1,t}, .., Q_n^{d+1,t}\}$ to collect more information based on $Q_i^{d,t}$, $C$, and

---

**Algorithm 2:** SELF-QUESTIONING

---

**Input:** Question $Q_i^{d,t}$, Hint $H_i^{d,t}$, Context $C$, Current
      Depth $d$, Max Depth $d_m$, Current Turn $t$, Max
      Turn $t_m$, Question Answer Prompt $P_{QA}$,
      Question Generate Prompt $P_{QG}$, QA to Hint
      Prompt $P_{\text{QA2H}}$
**Output:** $< \mathcal{Q}^{d+1,t}, \mathcal{H}^{d+1,t}, C >$

1  Must_Answer ←**False**;
2  **if** $d = d_m$ **or** $t = t_m$ **then**
3    |   Must_Answer ←**True**;
    // call the Question-Answering module
4  $< A_i^{d,t}, confidence > \leftarrow$
    $QA(Q_i^{d,t}, H_i^{d,t}, C, P_{QA})$ ;
5  **if** $confidence = high$ **or** $Must\_Answer$ **then**
6    |   **if** $d \neq 0$ **then**
        |   // merge QA to a hint
7          |   $\widetilde{H}_i^{d,t} \leftarrow QA2H(Q_i^{d,t}, A_i^{d,t}, P_{\text{QA2H}})$ ;
8    |   **else**
9          |_   $\widetilde{H}_i^{d,t} \leftarrow A^d$ ;
10   |   $\mathcal{Q}^{d+1} \leftarrow \emptyset$;
11   |   $\mathcal{H}^{d+1,t} \leftarrow \{\widetilde{H}_i^{d,t}\}$;
12 **else**
    |   // call the Question-Generation
    |     module
13   |   $\mathcal{Q}^{d+1,t} \leftarrow QG(Q_i^{d,t}, H_i^{d,t}, C, P_{QG})$ ;
14   |   $\mathcal{H}^{d+1,t} \leftarrow \emptyset$;
15 **return** $< \mathcal{Q}^{d+1,t}, \mathcal{H}^{d+1,t}, C >$;

---

$H_i^{d,t}$, where $n < n_m$ and $n_m$ denotes the maximum number of sub-questions to be generated. Algorithm 2 shows the pseudo-code of the SELF-QUESTIONING algorithm.

### 3.2.1 Question-Answering (QA) Module

The QA module aims to answer either the target question or a sub-question asked by the SELF-QUESTIONING module, based on the optional context and hints. We propose to leverage a large-scale language model (LLM), such as GPT-3 or Chat-GPT (OpenAI, 2022), to answer the question given their superior reasoning capabilities demonstrated in previous studies (Brown et al., 2020; Zhang et al., 2022; Wei et al., 2022; Touvron et al., 2023; Yao et al., 2023).

Specifically, the input to the QA module consists of the given question $Q_i^{d,t}$, the context $C$, the optional hints $H_i^{d,t}$, and a prompt $P_{QA}$ designed to guide the QA module to generate an answer $A_i^{d,t}$ based on the inputs and output a confidence level. When the hints $H_i^{d,t}$ are available, $P_{QA}$ also asks the QA module to indicate which hints ared used to produce the answer.

$$A_i^{d,t}, confidence = \text{QA}(Q_i^{d,t}, H_i^{d,t}, C, P_{QA}), \quad (1)$$

where $confidence \in \{high, medium, low\}$.

### 3.2.2 Question-Generation (QG) Module

When the QA module outputs an answer for question $Q_i^{d,t}$ with low confidence, it's very likely that the answer is not correct and we need to collect additional hints to help the QA module produce a more confident answer. To do so, we design a Question-Generation (QG) module to raise a set of sub-questions that are related to $Q_i^{d,t}$. The QG module is also based on a large language model, such as ChatGPT, that takes the question $Q_i^{d,t}$, optional hints $H_i^{d,t}$, the context $C$, and a prompt $P_{\text{QG}}$ as input and outputs a set of sub-questions:

$$\{\mathcal{Q}_0^{d+1}, ..., \mathcal{Q}_n^{d+1}\} = \text{QG}(Q_i^{d,t}, H_i^{d,t}, C, P_{QG}), \quad (2)$$

where $n < n_m$. Intuitively, the sub-questions should be simpler than $Q_i^{d,t}$ and more likely to be answered by the QA module with high confidence.

### 3.2.3 QA-to-Hint (QA2H) Module

Since the answers to sub-questions may not be self-contained, we further design a QA-to-Hint module (QA2H) to merge each sub-question with its answer into a statement. Specifically, we feed the sub-question $Q_i^{d,t}$ and its answer $A_i^{d,t}$ to an LLM with the prompt $P_{\text{QA2H}}$ which asks the LLM to rewrite the question to a statement by incorporating the answer:

$$\tilde{H}^d = QA2H(Q_i^{d,t}, A_i^{d,t}, P_{\text{QA2H}}), \quad (3)$$

## 4 SOCRATIC QUESTIONING for Few-Shot Multimodal Reasoning

SOCRATIC QUESTIONING can be naturally applied to text-based complex reasoning tasks as all the key components are based on large language models, such as ChatGPT. There are two critical challenges when applying SOCRATIC QUESTIONING to multimodal reasoning: (1) the language model cannot process visual information, and (2) simply applying a generic captioning model to convert visual content to natural language may not capture the key information required to answer a question.

**Converting Visual Information into Context** We propose to leverage LLMs to answer visual questions since some of the visual questions are knowledge-demanding (Marino et al., 2019; Schwenk et al., 2022) and LLMs are capable of storing commonsense knowledge and excel in complex reasoning tasks (Brown et al., 2020; Wei et al., 2022; Wang et al., 2023). To overcome the LLMs'

shortcomings that they cannot perceive visual information, previous works (Yang et al., 2022; Zeng et al., 2022) leverage an image captioning model to convert visual information into text and use LLMs to perform few-shot visual question answering (VQA) tasks. However, considering the richness and density of the information contained in an image, a generic caption may not be able to capture the key information that is necessary to answer a question. Thus, in order to adapt our SOCRATIC QUESTIONING, we employ a visual perception model, BLIP-2 (Li et al., 2023), to describe the content of the image that is specific to a prompt. The input to BLIP-2 is an image $I$ (i.e., the image input of the VQA task) and a text prompt $Q$, and the output is an image caption $C$ describing the part of the image related to the prompt: $C = \text{BLIP-2}(I, Q)$, where the text prompt $Q$ corresponds to $Q^d$ in Equation (1) and the caption $C$ corresponds to the context $C$ in Equation (1). By leveraging the visual perception model, we are able to resolve the hindrance and adopt our SOCRATIC QUESTIONING framework on VQA. We show more details on how we adapt SOCRATIC QUESTIONING to VQA in Appendix A.

## 5 Experiment Setups

**Language-Only Tasks** We leverage ChatGPT as the LLM for QA, QG, and QA2H modules, and provide detailed prompts for each module in Appendix K. We evaluate SOCRATIC QUESTIONING on several complex reasoning tasks, including the Physics and Chemistry tasks in **Massive Multitask Language Understanding (MMLU) (Hendrycks et al., 2020)**, Mathematical tasks in MATH (Hendrycks et al., 2021), and logical reasoning tasks based on LogiQA (Liu et al., 2020). We adopt several state-of-the-art prompting methods as baselines, including **Standard Prompting (SP)** that directly prompts ChatGPT to answers a question with a few in-context examples. **Chain-of-Thought (CoT) (Wei et al., 2022)**, **Self-Consistency Chain-of-Thought (SC-CoT) (Wang et al., 2023)**, and **Tree-of-Thought (ToT) (Yao et al., 2023)**. Following previous studies (Chowdhery et al., 2023; Hoffmann et al., 2022), we use exact match to measure the accuracy for all language-only tasks. More details for the baselines, evaluation metrics, and evaluation datasets are discussed in Appendix C.1.

**Multimodal Tasks** We use blip2-flan-t5-xl as our Visual Perception module. We leverage Chat-GPT (OpenAI, 2022) for Factual/Visual Question Generation and Factual Question Answering and GPT-3 (GPT-3-davinci-003) for Visual Question Answering[3], motivated by the observation that ChatGPT tends to be excessively cautious and neutral, and avoids answering some questions. We provide detailed sample prompts for each module in Appendix K. We evaluate SOCRATIC QUESTIONING on several visual question answering datasets, including **VQA-V2 (Goyal et al., 2017)**, **OK-VQA (Marino et al., 2019)** and **AOK-VQA (Schwenk et al., 2022)**, and compare our approach with several baselines, including **BLIP-2 (Li et al., 2023)** and. **PICa (Yang et al., 2022)**. More details for implementation, baselines, and datasets are discussed in Appendix C.2. For evaluation, we employ the conventional VQA accuracy metric (Goyal et al., 2017) to measure the performance. To alleviate stringent penalization for minor discrepancies between predicted answers and ground truth, we normalize the answers by converting plural forms to singular forms and changing the tense of verbs to present tense. In addition, to address the conventional metric's limitation due to synonyms and expression differences, we design semantic-based accuracy by employing Chat-GPT to evaluate the correctness of the predicted answers (Fu et al., 2023; Liu et al., 2023b). We provide ChatGPT with the visual question, the predicted answer and the ground-truth answer, and ask if the ground-truth answer and the predicted answer can support each other. If the answer is "Yes", we treat the predicted answer as correct. We show the exact prompts used for ChatGPT in Appendix K.8.

## 6 Results and Discussions

### 6.1 Quantitative Results

**Language-only Tasks** Table 1 shows the quantitative results in terms of accuracy for language-only reasoning tasks. Our method substantially outperforms previous state-of-the-art methods by 4.34%, 2.98%, 4.22%, and 4.66% absolute gains in MATH, Physics, Chemistry, and Logic benchmarks, respectively. This effectively demonstrates the superiority of our approach. We also conduct an experiment on how the maximum number of turns $t_m$ affects the performance. Specifically, we experiment with the setting where $t_m = 2$ (2-Turns)

---

[3] These components are detailed in Appendix A.

|  | MATH (DA) | MMLU Physics | MMLU Chemistry | LogiQA | Avg |
|---|---|---|---|---|---|
| Standard-Prompting | 7.00 | 65.11 | 53.20 | 54.67 | 45.00 |
| CoT (Wei et al., 2022) | 7.33 | 67.66 | 57.14 | 48.33 | 45.12 |
| SC-CoT (Wei et al., 2022) | 7.00 | 68.51 | 59.33 | 49.00 | 46.03 |
| ToT (Yao et al., 2023) | 0.00 | 40.00 | 26.60 | 22.22 | 29.46 |
| SOCRATIC QUESTIONING (2-Turns) | 7.67 | **71.49** | 63.55 | **59.33** | 50.51 |
| SOCRATIC QUESTIONING (3-Turns) | **11.67** | 69.36 | **63.55** | 58.00 | **50.65** |

Table 1: Accuracy (%) using Exact Match. The best performance is highlighted in **bold** and the second best performance is highlighted with underline.

> **Question**: A heavy rock and a light rock in free fall (zero air resistance) have the same acceleration. The heavy rock doesn't have a greater acceleration because the
> **Option**: ["A. force due to gravity is the same on each.", "B. air resistance is always zero in free fall.", "C. inertia of both rocks is the same.", "D. ratio of force to mass is the same."]

**Chain-of-Thought**

The force of gravity depends on the mass of each object and the distance between them, but since both objects are in free fall, they are at the same distance from the center of the Earth and experience the same force of gravity. Thus, option A, "force due to gravity is the same on each," is the correct answer. Thus, the final answer is: A.

**Tree-of-Thought**

To answer the question, we can use Newton's second law of motion, which states that force equals mass times acceleration. Since both rocks are in free fall, they are only under the influence of gravity, which means that the force due to gravity is the same on each rock. Therefore, option A is the correct answer.

**Socratic Questioning:**

- Hints:
    1. The acceleration due to gravity is 9.81 m/s^2.
    2. The formula for force due to gravity is F = (G*m1*m2) / r^2.
    3. The mass of an object does not affect its acceleration in free fall.
- Answer:
By hints 1 and 3, the heavy rock does not have a greater acceleration than the light rock because their force to mass ratio is the same. Option A is incorrect because according to hint 2, the heavy rock experiences a greater gravitational force due to its larger mass. Option B is incorrect because air resistance is not a factor for objects in free fall. Option C is incorrect because the inertia of an object is proportional to its mass, so the heavy rock has greater inertia than the light rock. Thus, the final answer is: D

Figure 4: Qualitative results of CoT, ToT, and SOCRATIC QUESTIONING on the Physics task. The correct answer of this example is **D**.

| Model | VQA-V2 | OK-VQA | AOK-VQA |
|---|---|---|---|
| BLIP-2 (Li et al., 2023) | 36.7 | 21.14 | 0 |
| PICa (Yang et al., 2022) | 43.18 | 29.94 | 28.6 |
| SOCRATIC QUESTIONING | **46.64** | **31.24** | **29.58** |

Table 2: Traditional VQA Accuracy (%) based on Exactly Match. The best performance is highlighted in **bold** and the second best performance is highlighted with underline.

| Model | VQA-V2 | OK-VQA | AOK-VQA |
|---|---|---|---|
| BLIP-2 (Li et al., 2023) | **57.2** | 46.75 | 43.29 |
| PICa (Yang et al., 2022) | 49.8 | 48.05 | 46.85 |
| SOCRATIC QUESTIONING | 54.4 | **53.03** | **49.55** |

Table 3: Semantic-based VQA Accuracy (%) using NLI. The best performance is highlighted in **bold** and the second best performance is highlighted with underline.

and $t_m = 3$ (3-Turns). From Table 1, the model with maximum 2 turns achieves better performance on Physics and LogiQA datasets, while the model with $t_m = 3$ performs better on the MATH dataset. One possible reason is that the Physics and LogiQA benchmarks may not be challenging enough and reasoning within 2 turns is sufficient to answer most of the questions. We provide a concrete example in Appendix G.1.

**Multimodal Tasks** Table 2 and 3 show the quantitative results using traditional VQA accuracy and semantic-based accuracy, respectively. For both results, our SOCRATIC QUESTIONING method outperforms the previous state-of-the-art approaches

on most benchmarks, often by a large margin. The only exception is semantic-based accuracy on the VQA-V2 dataset. A possible reason is that the tasks on VQA-V2 focus more on the visual recognition and detection aspect and do not require much reasoning capability and external knowledge.

### 6.2 Qualitative Result

**Language-only Tasks** Figure 4 shows the qualitative results of SOCRATIC QUESTIONING and baselines on the Physics task. As one can observe, SOCRATIC QUESTIONING can effectively prompt hints containing the necessary information to solve the original problem and selectively use the hints to reach the correct final answer. On the other hand, CoT and ToT reach the wrong answer due to the

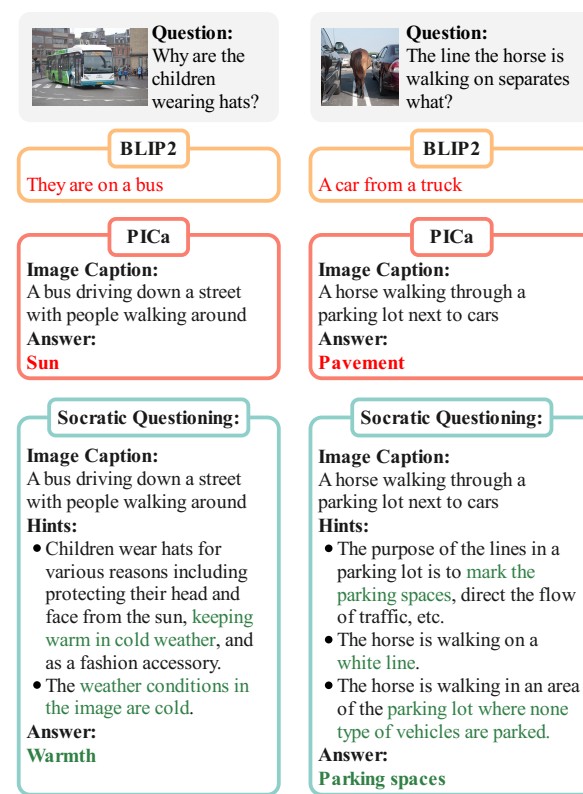

Figure 5: Qualitative results of few-shot VQA using BLIP-2, PICa, and SOCRATIC QUESTIONING (2-Depth 2-Turn).

|  | Answered Correctly | Answered Incorrectly |
|---|---|---|
| Avg. Hints | 3.28 | 3.68 |
| Avg. Depth | 2.89 | 2.92 |

Table 4: Averaged numbers of hints and depth of SO-CRATIC QUESTIONING used for questions answered correctly and incorrectly, respectively.

poorly sampled reasoning path.

**Multimodal Tasks**  Figure 5 shows several examples of few-shot VQA tasks from the baselines and SOCRATIC QUESTIONING. We demonstrate that the hints acquired via the sub-problems are highly related to the original problem (e.g., *"weather conditions are cold"*), and by considering the collected hints, the SOCRATIC QUESTIONING reaches the correct final answer (e.g., *"warmth"*). In contrast, the answer from BLIP-2 is irrelevant to the given question, due to the generic caption.

### 6.3 How do the Numbers of Turns and Depths Affect the Model?

**Performance Breakdown w/ Number of Turns**
To study how the number of reasoning turns affects the performance across different benchmarks, we investigate how the baselines and our method per-

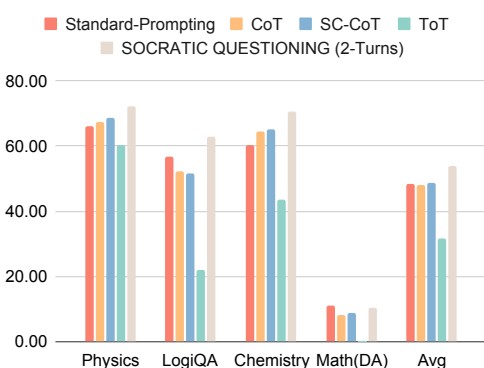

Figure 6: Accuracy (%) on the examples that triggered 2 turns of reasoning by SOCRATIC QUESTIONING.

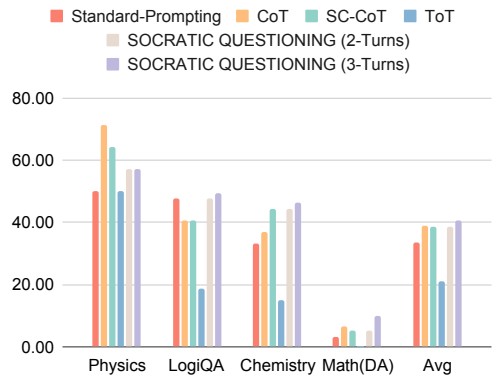

Figure 7: Accuracy (%) on the examples that triggered 3 turns of reasoning by SOCRATIC QUESTIONING.

form on the examples that triggered 2 and 3 turns of reasoning by SOCRATIC QUESTIONING in Figure 6 and Figure 7, respectively. This experiment can be considered as breaking down the results in Table 1 into two groups based on the number of reasoning turns. From Figure 6, our approach outperforms the baselines on all benchmarks except for the MATH dataset. From Figure 7, our approach outperforms the baselines on relatively challenging tasks such as MATH but performs more poorly on easier tasks such as Physics. This indicates SOCRATIC QUES-TIONING with more turns can tackle challenging problems more effectively.

**The Effect of Hyperparameters** $t_m$ **and** $d_m$  In addition to the discussion in 6.1, we conduct a more in-depth analysis of how the maximum number of turns $t_m$ and maximum number of depths $d_m$ affect the performance of our SOCRATIC QUESTIONING. In Figure 8, we show the heat map under different hyperparameter settings, where the number in each cell is the accuracy (%) given a specific combination of $t_m$ and $d_m$. We observe two general trends: (1) the accuracy increases when $t_m$ gets larger, and

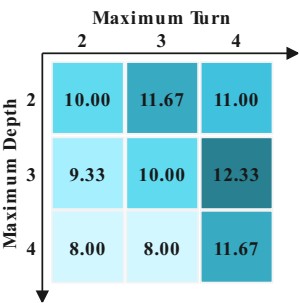

Figure 8: Quantitative results SOCRATIC QUESTION-ING on the MATH dataset with different values of the hyperparameters $t_m$ and $d_m$.

(2) the accuracy decreases when $d_m$ gets larger. These results imply that our approach can benefit from raising more questions directly related to the original question. Also, performing reasoning with a larger maximum depth does not yield better performance since the benchmark may not be challenging enough, and exploring at a deeper level may introduce irrelevant information. We provide a concrete example in Appendix G.2. In addition, we analyze the computational cost of SOCRATIC QUESTIONING compared to other baselines in Appendix H, and show that while achieving stronger performance, our proposed algorithm enjoys higher efficiency than most of baselines.

### 6.4 How does the Difficulty of Questions Affect the Model?

Table 4 presents the averaged numbers of hints and depth used to answer the original questions for correct and incorrect answers. As one can observe, for incorrect answers, the LLM raises more sub-questions, which demonstrates that the LLM tends to explore more thinking space when tackling questions that it does not know the answers. This trend also agrees with the depth. If the question is hard for the LLM, the model tends to break the sub-questions into even more basic questions.

## 7 Conclusion

We present SOCRATIC QUESTIONING, a novel divide-and-conquer fashion algorithm that is inspired by human's recursive thinking processes. SOCRATIC QUESTIONING consists of a top-down reasoning phase that decomposes a complex problem into simpler sub-problems and a bottom-top phase where the solutions to the sub-problems are recursively returned and used to solve the original problem at higher levels. Extensive experiments

on four challenging language-only tasks and the few-shot VQA task validate the effectiveness of our SOCRATIC QUESTIONING. Moreover, qualitative analysis demonstrates our approach can effectively elicit intermediate reasoning steps and consequently yield a correct final answer while enjoying transparency and interpretability.

## Limitation

The self-checking functionality lacks sufficient sensitivity to incorrect responses, as its confidence estimation heavily relies on LLMs themselves. While we employed ChatGPT as the backbone for our algorithm, its tendency towards overconfidence leads to a low frequency of sub-question generation.

Our study exhibits a lack of diversity in visual models used to extract information from images. We only use BLIP-2 (Li et al., 2023) as an image caption model in current experiments. However, the incorporation of diverse visual models, such as dense caption models, Optical Character Recognition (OCR), or scene graph models, may potentially yield a broader spectrum of image information, thus facilitating the resolution of sub-questions. In addition, to help BLIP-2 to better follow instructions from LLMs, we propose to leverage recent techniques developed in visual instruction tuning (Liu et al., 2023a; Xu et al., 2023b,a; Dai et al., 2023).

Additionally, our experiments were constrained to the English language datasets and we only consider the VQA task to showcase the multi-modal performance. However, given the generality of our algorithm, we plan to test its functionality with multilingual datasets and experiment it on other domains, such as speech (You et al., 2020, 2022), and video (Rose et al., 2023).

## Acknowledgments

This research is based upon work supported by the U.S. DARPA ECOLE Program # HR001122S0052. The views and conclusions contained herein are those of the authors and should not be interpreted as necessarily representing the official policies, either expressed or implied, of DARPA or the U.S. Government. The U.S. Government is authorized to reproduce and distribute reprints for governmental purposes notwithstanding any copyright annotation therein.

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

## A  Adapting SOCRATIC QUESTIONING to Visual Question Answering

**Question-Generation (QG) Module**  Some tasks (e.g., OK-VQA, AOK-VQA) require commonsense knowledge. Although LLMs can retrieve knowledge from its parameter, they are prone to hallucination and the black-box retrieving process is hard to debug. In order to gain a clear understanding of the factual knowledge used in answering a question, we divide the QG module in Section 3.2.2 into two sub-modules: A Fact-Question-Generation (FQG) sub-module which generates factual questions related to background knowledge of the given question, and a Visual-Question-Generation (VQG) sub-module generates visual questions, which aims to guide the Visual Perception module to focus on question-related image regions and seek more image information.

**Question-Answering (QA) Module**  To accommodate the two question types, we also divide the QA module in section 3.2.1 into two sub-modules: A Factual-Question-Answering module (FQA) and a Visual-Question-Answering module (VQA). Both FQA and VQA modules follow the same formulation in Equation (1). The input $C$ to VQA is the caption related to the question $Q^d$ and is prompted via the Equation of BLIP-2.

**SELF-QUESTIONING**  Figure 9 demonstrates the detailed step of the SELF-QUESTIONING algorithm in the multimodal setting. At depth $d$, SELF-QUESTIONING algorithm takes in a visual question $Q^d$ which can be the original visual question ($d = 0$) or a sub-question generated by VQG, a question-related caption $C$, and hints $H^d$ (if it is available), and try to generate an answer $A^d$ via VQA. If the confidence level of $A^d$ is not high, the SELF-QUESTIONING algorithm starts to raise sub-questions. First, the FQG module takes in $Q^d$, context $C$, and hints $H^d$ as input and raises a set of factual questions $\mathcal{Q}_f$. Each question in $\mathcal{Q}_f$ is answered by the FQA module and we denote the answer as $A_f$. Each $Q_f$ and its answer $A_f$ is mearged into a factual statement $h_f$ via the QA2H module

and the statement is appended to hints $H^d$ to form $H^{d+1}$. Second, the VQG module takes in $Q^d$, context $C$, and hints $H^{d+1}$ and raises a set of visual questions $Q^{d+1}$.

## B  Visualization of Recursive Thinking Process

Figure 10 shows a complete recursive thinking process of our SOCRATIC QUESTIONING method. It involves 4 additional questions to acquire additional information to answer the target question. From this example, we see that LLMs, such as GPT-3 or ChatGPT, have strong capabilities not only in reasoning but also self-questioning. Given the target question to be answered, "*Why are the children wearing hats?*", LLMs are able to proactively acquire additional commonsense knowledge through factual questions, e.g., "*What are the common reasons why children wear hats?*", and fine-grained visual information from the input image, e.g., "*What's the position of the sun in the sky at the time the children are shown wearing hats*", "*Are the weather conditions in the image cold or hot*". By combining the additional knowledge, e.g., "*cold weather makes people wear hats*" and visual information, e.g., "*it is cold*", acquired from the recursive Self-Questioning process, the model finally achieves the answer "*warmth*". This analysis demonstrates that the recursive thinking process of our approach is highly transparent and interpretable.

## C  Implementation Details

### C.1  Language-only Tasks

**Implementation Details**  We leverage Chat-GPT (OpenAI, 2022) as the LLM for QA, QG, and QA2H modules. We provide detailed prompts for each module in Appendix K.

**Baselines  Standard Prompting (SP)** prompts ChatGPT to directly answers a question with a few in-context examples. **Chain-of-Thought (CoT) (Wei et al., 2022)** prompts ChatGPT to first generate the thinking process and then generate the answer. We also add the thinking process into the in-context examples. **Self-Consistency Chain-of-Thought (SC-CoT) (Wang et al., 2023)** proposes to run chain-of-thought multiple times on Chat-GPT and marginalize the thinking process by taking the most consistent answer. **Tree-of-Thought (ToT) (Yao et al., 2023)** is a recently proposed

framework for improving the reasoning capability of language models. We follow their implementation [4] which leverages tree-search algorithms to explore the thinking space and select the best thinking path. [5]

**Evaluation Metrics**  For a fair comparison, we use exact match and measure the accuracy for all language-only tasks following previous works (Chowdhery et al., 2023; Hoffmann et al., 2022).

All questions in MMLU Physics, MMLU Chemistry, and LogiQA are multiple-choice questions and the answer is always a single letter like "A", "B" or "C". To easily parse the model's final output, we use "Thus, the final answer is:" as the prefix for the final answers (A or B or C or D, ect.) in the in-context examples for all methods. When we parse the output, we first run a template-based method to extract the answers after "Thus, the final answer is:". For a few instances (12.52% in CoT, 16.4% in ToT and 11.64% in Socratic Questioning on average) that do not match the template as shown in Figure 4 ToT, the authors manually compare the model's predictions to the ground truth answers. Thus, we assure that the final performance of all methods is not affected by the output formats.

**Datasets  Massive Multitask Language Understanding (MMLU) (Hendrycks et al., 2020)** dataset contains 57 diverse tasks and is used to measure the model's complex reasoning capability. In this work, we use the physics and chemistry tasks which contain conceptual physics and chemistry multiple-choice questions, respectively. **MATH (Hendrycks et al., 2021)** dataset consists of challenging competition-level mathematics problems which require strong mathematical reasoning ability. **LogiQA (Liu et al., 2020)** dataset contains expert-written questions for testing the logical reasoning capability of humans. For each task, we use the validation set to make design decisions and measure the model's performance on the test set. The detailed statistics of all datasets can be found in Table 5.

---

[4] https://github.com/kyegomez/tree-of-thoughts
[5] By the time we submit the work, we don't have access to GPT4 so we use ChatGPT for ToT.

|       | MATH | MMLU (Physics) | MMLU (Chemistry) | LogiQA |
|-------|------|----------------|------------------|--------|
| **Dev** | 60   | 22             | 26               | 60     |
| **Test** | 300  | 235            | 203              | 300    |

Table 5: Statistic of datasets for language-only tasks.

## C.2 Multimodal Tasks

**Implementation Details** We use blip2-flan-t5-xl[6] as our Visual Perception module. We leverage ChatGPT (OpenAI, 2022) for the FQG, VQG, and FQA modules and GPT-3 (GPT-3-davinci-003) for the VQA module. This decision is motivated by the observation that ChatGPT tends to be excessively cautious and neutral, and avoids answering some questions. We provide detailed sample prompts for each module in Appendix K.

**Baselines** BLIP-2 (Li et al., 2023) is a pre-trained vision-language model that leverages an efficient and generic pre-training strategy and is able to follow text prompts. We use the released blip2-flan-t5-xl checkpoint. **PICa (Yang et al., 2022)** prompts GPT-3 with generic image captions to solve VQA in an in-context learning manner. In our experiments, we implement PICa by using blip2-flan-t5-xl as the image captioning model and `GPT-3-davinci-003` as the LLM.

**Evaluation Metrics** We employ the conventional VQA accuracy metric (Goyal et al., 2017) to measure the performance. To alleviate stringent penalization for minor discrepancies between predicted answers and ground truth, we normalize the answers by converting plural forms to singular forms and changing the tense of verbs to present tense. In addition, to address the limitation due to synonyms and expression differences, we employ Chat-GPT to evaluate the correctness of the predicted answers (Fu et al., 2023; Liu et al., 2023b). We provide ChatGPT with the visual question, the predicted answer and the ground-truth answer, and ask if the ground-truth answer and the predicted answer can support each other. If the answer is "Yes", we treat the predicted answer as correct. We show the exact prompts used for ChatGPT in Appendix K.8.

**Datasets** VQA-V2 (Goyal et al., 2017) is a dataset containing open-ended questions about images. **OK-VQA (Marino et al., 2019)** requires

[6]https://huggingface.co/Salesforce/blip2-flan-t5-xl

|       | VQA-v2 | OK-VQA | AOK-VQA |
|-------|--------|--------|---------|
| **Dev** | 100    | 100    | 100     |
| **Test** | 500    | 462    | 444     |

Table 6: Statistic of datasets for multi-modalities tasks.

model to leverage external knowledge to answer visual questions. **AOK-VQA (Schwenk et al., 2022)** is an augmented successor of OK-VQA, which require commonsense knowledge and strong reasoning capabilities to answer its questions. For each task, we use the validation set to make design decisions and measure the model's performance on the test set. The detailed statistics of all datasets can be found in Table 6 and Appendix E.

## D SELF-QUESTIONING in the Multimodal Setting

See Figure 9.

## E Data Leakage in BLIP-2 and GPT-3

| Model  | VQA-V2 | OK-VQA | AOK-VQA |
|--------|--------|--------|---------|
| BLIP-2 | 1.46   | 2.93   | 28.08   |
| GPT-3  | 35.88  | 23.95  | 20.4    |

Table 7: Traditional VQA Accuracy (%) under the setting where no image is provided in the input.

In our preliminary experiments, we discovered an issue that pre-trained models could be subject to data leakage during their pre-training stage. We observed that the baseline models (i.e., BLIP-2 and GPT-3) achieved unjustifiably high performance across all three VQA datasets even without taking images as inputs (see Table 7). To address this issue, we applied a filtering process to remove such contaminated instances. We first test the BLIP-2 and GPT-3 on zero-shot VQA tasks while replacing the original input image with an image composed entirely of black pixels of the same size. Then, we only retain the samples where the models failed to yield a correct answer when the original image is not given. After the filtering, we adopt the 500, 462, and 444 test samples for VQA-V2, OK-VQA, and AOK-VQA, respectively. We use these clean examples for the evaluation throughout the rest of our experiments.

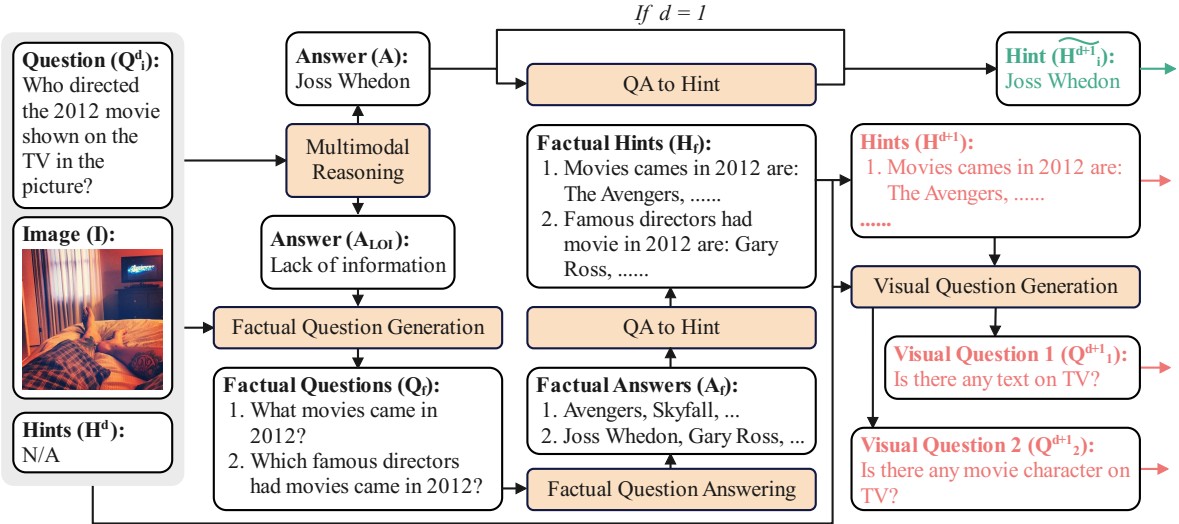

Figure 9: The overview of the SELF-QUESTIONING Algorithm.

## F   Visualization of Complete SOCRATIC QUESTIONING

See Figure 10.

## G   Concrete Example

### G.1   Large Maximum Number of Turn

Due to the calibration error in LLMs (Jiang et al., 2021), sometimes the pre-trained model's confidence is not aligned with the answer's correctness. Thus, in such cases, the model predicts "low" or "medium" confidence in correct answers in the early turns and hence misses the correct answers. If we use fewer turns, we can keep the answer in the early turn regardless of the confidence and hence alleviate the calibration error. Below we show a concrete example in which the model predicts the correct answer in 2 turns and predicts the incorrect answer in 3 turns. When we increase the number of turns, Socratic Questioning may raise some less relevant sub-questions and hence introduce noisy information in the reasoning process. This noisy information can confuse the model, leading to incorrect responses to the original question. For example, consider a simple physics question:

*The speed of sound is slightly greater on a [ "A. cold day", "B. hot day", "C. day with steady temperature", "D. None of these"]?*

In a 2-turn setting, our approach obtains hints: (1) "The speed of sound increases with increasing temperature.", and (2) "Humidity is a factor in the speed of sound." According to the hints, it is obvious that the correct answer is B, which is chosen by

our approach in the second turn with the "middle" confidence. In a 3-turn setting, since the LLM does not assign "high" confidence to the answer in the 2 turn, our approach goes deeper in the third turn and gets more information (e.g., (3) "The speed of sound can be affected by several factors, including temperature, humidity and density of the medium.", (4) "The speed of sound depends on the density and elasticity of the medium it is traveling through, in terms of physical properties.", (5) "The speed of sound increases with humidity as a result of increased air density.") As a result, by considering more hints, we potentially introduce less relevant information to the LLM and the noisy information causes the LLM to change its answer to D.

### G.2   Large Maximum Number of Depth

We observe that as the depth increases, the context information in the original questions start to vanish and the answers to the sub-questions may be inaccurate in the context of the original question. Thus, by adding the answers to sub-question in larger depth as hints, we can introduce noises to the reasoning process of the LLM which results in wrong answers. Consider a physics question example:

*When a spinning system contracts in the absence of an external torque, its rotational speed increases, and its angular momentum [ A. decreases, B. increases, C. remains unchanged, D. may increase or decrease ]"?*

Socratic Questioning raises a sub-question: *"What affects the rotational speed of a spinning system?"* The initial answer to this sub-question

is *"Conservation of angular momentum"*, which provides enough information to answer the original question. In a larger depth setting, the Socratic Questioning raises a deeper sub-question: *"What is the relationship between rotational speed and angular momentum in a spinning system?"* The answer to this question is: *"The angular momentum is directly proportional to the rotational speed"*. Incorporate this hint, the Socratic Questioning changes the answer of the first sub-question to: *"The angular momentum is directly proportional to the rotational speed."*, which results in an incorrect final answer B.

## H    Evaluation of Computational Cost

In Table 8, we provide the theoretical number of calls in CoT, SC-CoT, ToT and Socratic Questioning in 2 and 3 turns settings. We also provide the empirical results of the average number of calls per instance and average running time per instance in seconds for all methods. For SC-CoT, we fix the number of calls to 20 times on all the datasets based on the performance curve in (Wang et al., 2023). In ToT, k represents the number of thoughts allowed to be generated per step, T represents the maximum number of steps and b represents the maximum number of states to keep at each step in BFS. Following (Yao et al., 2023), we set k=5, T=3, and b=4. In Socratic Questioning, q represents the maximum number of raised sub-questions for a parent node.

As one can observe, Socratic Questioning with 2 turns and 3 turns achieves better efficiency compared to SC-CoT and ToT. The main reason is that, in the experimental datasets, most questions do not require a large amount of thinking steps to reach the correct answers. Socratic Questioning, adaptively raises sub-questions based on the complexity of the original question and arrives at the correct answer without reaching the theoretical maximum number of turns or depth. In contrast, both SC-COT and ToT employ fixed settings for the number of thoughts generated per step. For relatively straightforward questions, these fixed settings introduce high computational overhead, making the algorithms less efficient in these questions.

## I    Experimental Results on Other QA and Math Datasets

Table 9 provides the performance of our method and two strong baselines on GSM8K and Strate-

gyQA datasets. As one can observe, our method has significant performance improvement compared to baselines. We use ChatGPT with temperature 0.7 for all methods. For SC-CoT, we sample 20 reasoning paths.

We tried our best to reproduce the results of CoT and SC-CoT reported in (Wang et al., 2023) on StrategyQA. Following (Wang et al., 2022), we use the question-only set from BIG-bench collaboration (2021) and use the exact same prompt template and in-context examples in SC-CoT. However, we cannot reproduce the results on StrategyQA in (Geva et al., 2021) since Code-davinci-002 and Code-davinci-001 are no longer publicly available. In addition, our results of ChatGPT on StrategyQA also agree with more recent studies in (Qin et al., 2023).

## J    Experiment Results based on GPT-4

To showcase the generalizability of our approach, we have run CoT and Socratic Questioning on MMLU Chemistry and LogiQA based on GPT-4. The experimental results show that our Socratic Questioning approach still significantly outperforms CoT.

## K    Prmopt Templates

To make our method generalize to other reasoning domains, we carefully design in-context demonstrations to guide the LLM to generate more basic sub-questions in an efficient manner. More concretely, to create high-quality sub-questions in the in-context examples, we take the human reasoning process and domain knowledge into account and carefully annotate the sub-questions by ensuring that they are more basic questions compared to the original question and their solutions can contribute to the reasoning process of the original questions. For examples of sub-questions, please refer to Figure 12. Based on our experiments in math, physics, chemistry and VQA domains, we argue that with a few examples (5 in all our experiments) Socratic-Questioning can generalize to a new domain.

Following (Kadavath et al., 2022), we ask the LLM itself to output a confidence level, "high", "middle", or "low", towards its answer. In the in-context demonstrations, we label the correct answers with supportive hints in the context as "high" confidence, label the correct answers without supportive hints as "middle" confidence, and label incorrect answers as "low" confidence. In this way,

| | Standard-Prompting | CoT | SC-CoT | ToT | Socratic Questioning (2 turns) | Socratic Questioning (3 turns) |
|---|---|---|---|---|---|---|
| Theoretical Number of Calls | 1 | 1 | 20 | k + b*k*(T-1) | $3 \times \sum_{i=1}^{d-1}[q \times (t-1)]^i$ | $3 \times \sum_{i=1}^{d-1}[q \times (t-1)]^i$ |
| Avg. Calls per Instance | 1 | 1 | 20 | 31.1 | 9.22 | 18.7 |
| Avg. Running Time per Instance (second) | 0.33 | 3.35 | 67.09 | 77.99 | 34.15 | 53.65 |

Table 8: Evaluation of computational cost of different methods.

| | GSM8K | StrategyQA |
|---|---|---|
| CoT | 79.0 | 59.7 |
| SC-CoT | 86.0 | 63.0 |
| Socratic-Questioning | **89.33** | **65.33** |

Table 9: Accuracy (%) on GSM8K and StrategyQA using Exact Match. The best performance is highlighted in **bold**.

| | MMLU Chemistry | LogiQA |
|---|---|---|
| CoT | 80.2 | 70.3 |
| Socratic-Questioning | **85.73** | **75.3** |

Table 10: Accuracy (%) of GPT-4 based approaches using Exact Match. The best performance is highlighted in **bold**.

we can guide the model to align its confidence to the correctness of the predicted answers. Our algorithm will continue raising sub-questions if the estimated confidence is not "high". Please refer to Figure 11 for more examples.

### K.1 Prompts template of QA module

See Figure 11.

### K.2 Prompts template of QG module

See Figure 12.

### K.3 Prompts template of FQG

See Figure 13.

### K.4 Prompts template of FQA

See Figure 14.

### K.5 Prompts template of VQG

See Figure 15.

### K.6 Prompts template of VQA

See Figure 16 and 17.

### K.7 Prompts template of QA-to-Hint

See Figure 18.

### K.8 Prompt for chatGPT for semantic-based accuracy evaluation

See Figure 19.

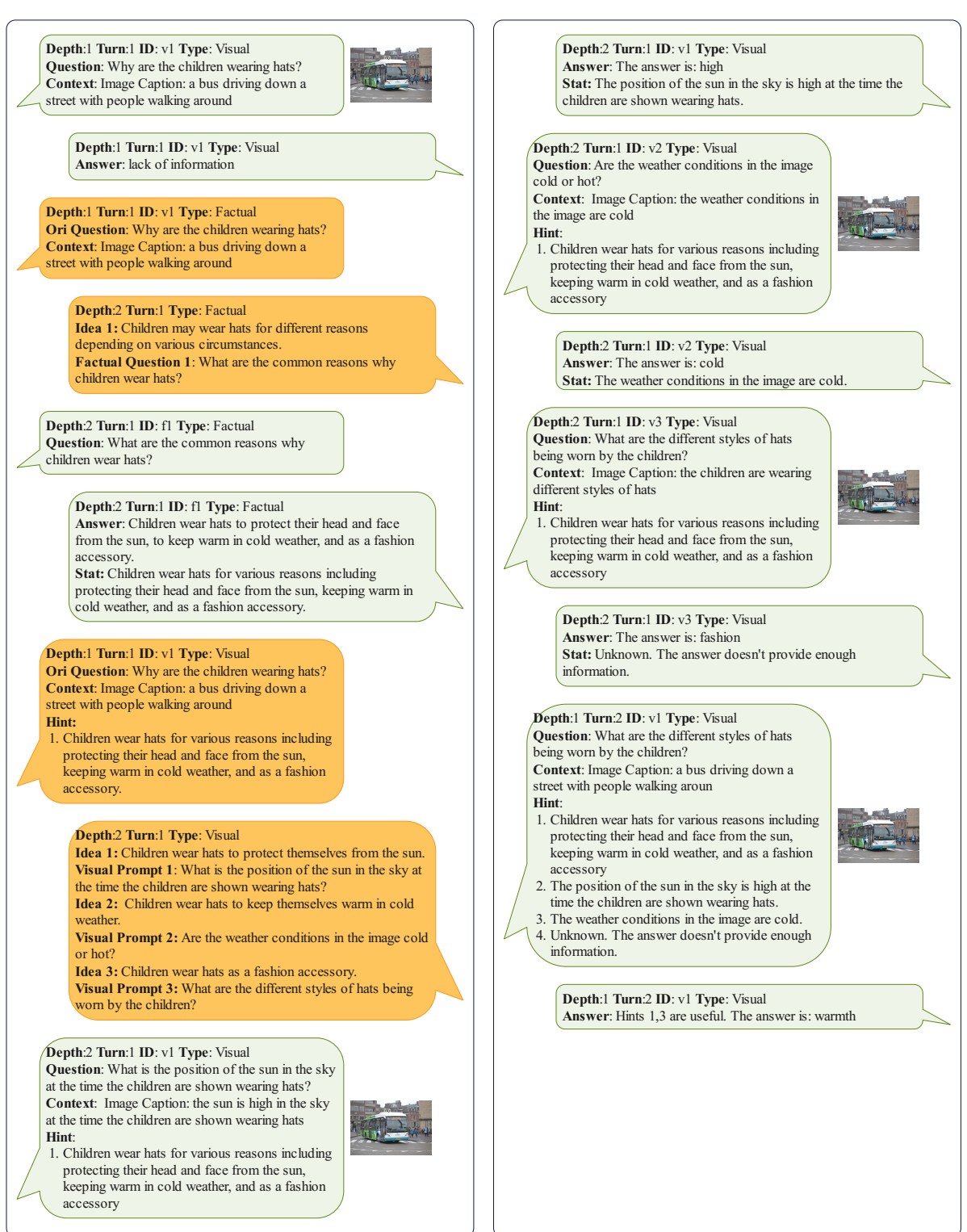

Figure 10: Visualization of a complete recursive thinking process of SOCRATIC QUESTIONING (2-Depth 2-Turn). The process is initialized on the left and is continued on the right.

// Demonstration

**Question**:
The plates of a capacitor are charged to a potential difference of 5 V. If the capacitance is 2 mF, what is the charge on the positive plate?

**Option**:
[A. 0.005 C, B. 0.01 C, C. 0.02 C, D. 0.5 C]
Note: Form of output is: explanation + [Answer: "option letter"; Used hints: "hints ID or None"; Confidence: "low, middle, or high"].

**Answer**:
To find the charge on the positive plate of the capacitor, we can use the formula Q = CV, where Q represents the charge, C represents the capacitance, and V represents the potential difference. Given that the potential difference V is 5 V and the capacitance C is 2 mF (millifarads), we can substitute these values into the formula to find the charge Q. Q = (2 mF) * (5 V) To simplify the calculation, we need to convert millifarads (mF) to farads (F). Since 1 mF is equal to 0.001 F, we can rewrite the capacitance as 2 * 0.001 F. Q = (2 * 0.001 F) * (5 V) = 0.01 C Thus, the charge on the positive plate of the capacitor is 0.01 C. Thus, the final answer is: [Answer: B; Used hints: None; Confidence: low]

**Question**:
How much work is required to charge a 10 μF capacitor to a potential difference of 100 V ?

**Option**:
[A. 0.005 J, B. 0.01 J, C. 0.05 J, D. 0.1 J]
Note: Form of output is: explanation + [Answer: "option letter"; Used hints: "hints ID or None"; Confidence: "low, middle, or high"].

**Answer**:
To calculate the work required to charge a capacitor, we can use the formula: Work = 0.5 * C * V^2, where C represents the capacitance and V represents the potential difference. Given that the capacitance C is 10 μF (microfarads) and the potential difference V is 100 V, we can substitute these values into the formula: Work = 0.5 * (10 μF) * (100 V)^2. To simplify the calculation, we need to convert microfarads (μF) to farads (F). Since 1 μF is equal to 1 × 10^-6 F, we can rewrite the capacitance as 10 × 10^-6 F. Work = 0.5 * (10 × 10^-6 F) * (100 V)^2 = 5 * 10^-5 F * 10^4 V^2 = 0.5 * 10^-1 F * 1 V^2 = 5 * 10^-2J. Thus, the final answer is: [Answer: C; Used hints: None; Confidence: low]

**Question**:
Ultraviolet light has a wavelength of about 6 × 10^-8 m. What is the frequency of this light?
Option: [A. 5 × 10^15 Hz, B. 0.5 Hz, C. 2 Hz, D. 20 Hz]

**Hints**:
(1) Wavelength is inversely proportional to the frequency.
Note: Form of output is: explanation + [Answer: "option letter"; Used hints: "hints ID or None"; Confidence: "low, middle, or high"].

**Answer**:
To find the frequency of ultraviolet light with a given wavelength, we can use the relationship between wavelength and frequency, which is hint 1: wavelength is inversely proportional to frequency. Mathematically, this relationship can be expressed as: frequency = speed of light / wavelength. The speed of light in a vacuum is approximately 3 × 10^8 meters per second (m/s). Given that the wavelength of the ultraviolet light is 6 × 10^-8 m, we can substitute these values into the formula: frequency = (3 × 10^8 m/s) / (6 × 10^-8 m) = 0.5 x 10^16 Hz= 5 x 10^15 Hz. Thus, the frequency of ultraviolet light with a wavelength of 6 × 10^-8 m is 0.5 Hz. Thus, the final answer is: [Answer: A; Used hints: 1; Confidence: high]

**Question**:
A whiffle ball is tossed straight up, reaches the highest point, and falls back down. Air resistance is not negligible. Which of the following statements are true? I. The ball's speed is zero at the highest point. II. The ball's acceleration is zero at the highest point. III. The ball takes a longer time to travel up to the highest point than to fall back down.

**Option**:
[A. I only, B. II only, C. I & II only, D. I & III only]
Note: Form of output is: explanation + [Answer: "option letter"; Used hints: "hints ID or None"; Confidence: "low, middle, or high"].

**Answer**:
Let's evaluate each statement one by one: I. The ball's speed is zero at the highest point. This statement is true. At the highest point of the ball's trajectory, its vertical velocity becomes zero. Since speed is the magnitude of velocity, which is a vector quantity, the speed of the ball indeed becomes zero at the highest point. This is because the ball momentarily stops moving upward before it starts descending. II. The ball's acceleration is zero at the highest point. This statement is false. The ball's acceleration is not zero at the highest point. Even though the ball momentarily stops changing its direction (from upward to downward motion) at the highest point, it still experiences the force of gravity acting downward. The presence of gravity causes the ball's acceleration to remain constant throughout its motion, regardless of the point in its trajectory. III. The ball takes a longer time to travel up to the highest point than to fall back down. This statement is false.

Due to the presence of air resistance, the ball experiences a drag force opposing its motion. As a result, the ball takes longer to reach the highest point of its trajectory compared to the time it takes to fall back down. Air resistance acts as a damping force, reducing the upward speed of the ball and increasing the time it takes to reach the peak. When falling back down, the ball's downward speed is increased by the force of gravity, making the descent faster than the ascent. Based on the evaluations, we can conclude that only Statement I is true. Thus, the final answer is: [Answer: A; Used hints: None; Confidence: middle]

**Question**:
An object of volume 2 × 10^-3 m^3 and weight 6 N is placed into a tank of water, where it floats. What percentage of the object's volume is above the surface of the water?

**Option**:
[A. 12%, B. 30%, C. 60%, D. 70%]

**Hints**:
(1) Density of water is 997 kg/m^3; (2) Object density is 306 kg/m^3.

**Note**:
Form of output is: explanation + [Answer: "option letter"; Used hints: "hints ID or None"; Confidence: "low, middle, or high"].

**Answer**:
The percentage of the object's volume that is above the surface of the water can be calculated using the formula: Percentage above surface = [(Density of water - Density of object) / Density of water] × 100. In this question, the percentage above surface = [(997 kg/m^3 - 306 kg/m^3) / 997 kg/m^3] × 100 = (691 kg/m^3 / 997kg/m^3) × 100 = 0.693 × 100 ≈ 70%. Thus, the final answer is: [Answer: D; Used hints: 1, 2; Confidence: high]

Figure 11: Prompt template of QA module.

Imagine you are a thoughtful and logical question-raiser. You are given a physics question. However, the question is too complex or lack of information to answer. You need to raise some questions to decompose the original question into several simpler sub-questions, or to seek additional information that helps you answer the original question. Important notes: do not use pronouns or indefinite pronoun phrases in your generated questions. The raised question has to be the self-contain question, which means including context if it is needed. Each question can only contain one argument. Do not just ask Yes/No questions.

*// Demonstration*

**Question**:
An object of volume $2 \times 10^{-3}$ m^3 and weight 6 N is placed into a tank of water, where it floats. What percentage of the object's volume is above the surface of the water?
**Option**:
[A. 12%, B. 30%, C. 60%, D. 70%]
**Note**:
The raised question has to be a self-contain question. Do not use pronouns or indefinite pronoun phrases in the generated questions. Copy context from the original question if needed.
**Deep Questions:**
1. When an object floats, what function describes the relationship between the object's volume and weight?
2. What is the density of water?
3. An object of volume $2 \times 10^{-3}$ m^3 and weight 6 N, what is the object's density?

**Question**:
Compared with the mass of a uranium atom undergoing fission, the combined masses of the products after fission are
**Option**:
[A. less, B. more, C. the same, D. zero]
**Note**:
The raised question has to be a self-contain question. Do not use pronouns or indefinite pronoun phrases in the generated questions. Copy context from the original question if needed.
**Deep Questions:**
1. What causes the change in mass of a particle before and after fission?

**Question**:
Things that are equivalent according to the equivalence principle are
**Option**:
[A. space and time, B. a traveling twin and a stay-at-home twin, C. gravity and acceleration, D. mass and energy]
**Note**:
The raised question has to be a self-contain question. Do not use pronouns or indefinite pronoun phrases in the generated questions. Copy context from the original question if needed.
**Deep Questions:**
1. What is the equivalence principle?

**Question**:
Which of these three elements has the most mass per nucleon?
**Option**:
[A. Hydrogen, B. Iron, C. Uranium, D. Same in each]
**Note**:
The raised question has to be a self-contain question. Do not use pronouns or indefinite pronoun phrases in the generated questions. Copy context from the original question if needed.
**Deep Questions:**
1. What is the nucleon mass of hydrogen?
2. What is the nucleon mass of iron?
3. What is the nucleon mass of uranium?

**Question**:
A microwave oven is connected to an outlet, 120000 mV, and draws a current of 2 amps. At what rate is energy being used by the microwave oven?
**Option**:
[A. 10 W, B. 30 W, C. 60 W, D. 240 W]
Note: The raised question has to be a self-contain question. Do not use pronouns or indefinite pronoun phrases in the generated questions. Copy context from the original question if needed.
**Deep Questions:**
1. Given a given voltage and current, how to calculate the power?
2. How many volts equal 12000 microvolts?

Figure 12: Prompt template of QG module.

Figure 13: Prompt template of FQG module.

Figure 14: Prompt template of FQA module.

Figure 15: Prompt template of VQG module.

Figure 16: Prompt template of VQA module.

Figure 17: Prompt template of VQA module (force answer).

```
// System define
You are given a question-and-answer pair, can you help
me to merge the question and answer into a statement
sentence. If the question or the answer is ambiguous you
can just output the token "unknown". If the merged
sentence is ambiguous, you can just output the token
"unknown". If you can merge the question-and-answer
pair, just output the sentence.

// Input
Question:
  • In which state in the USA are oranges grown?
Answer:
  • California
Statement:
```

Figure 18: Prompt template of QA-to-Hint module.

```
// System define
Imagine you are a strict marking teacher or grader. I will
give you a question, a correct answer, and a student
answer. You need to tell me "1" or "0" (where 1 means
correct, 0 means incorect). "1" does not mean the
student's answer must exactly match the correct answer.
If they have the same meaning for the given question,
then it is also "1". However, an ambiguous answer is "0"
(e.g., correct answer: "1789", student answer: "long long
ago").

// Input
Question:
  • In which state in the USA are oranges grown?
Correct Answer:
  • California
Student Answer:
  • California state
Grade:
```

Figure 19: Prompt template for chatGPT of semantic-
based evaluation.