# OpenReview forum: "The Art of SOCRATIC QUESTIONING: Recursive Thinking with Large Language Models"
_EMNLP/2023/Conference — EMNLP 2023 Main_

### Official Review · Reviewer_riYe · 2023-08-05

**Soundness:** 3

**Excitement:**

4: Strong: This paper deepens the understanding of some phenomenon or lowers the barriers to an existing research direction.

**Paper Topic And Main Contributions:**

This paper proposes an LLM inference pipeline called socratic questioning, which essentially is a question decomposition process, that decomposes the given question to simpler sub-questions if the current question cannot be confidently answered. The authors use a trick they call "hints" to feed the sub-question answers back to the original question by converting the sub-question and its answer into a statement. This paper also proposes an explainable VQA pipeline that is built on top of this socratic questioning method, which uses a novel trick that checks for image information with dynamic textual prompts, which are tailored to the need of specific sub-questions.

Experiments show that the proposed pipeline achieved superior performance on MATH, LogiQA, and MMLU, as well as VQA datasets in multimodal settings, compared with other prompting-based inference methods such as chain of thought.

**Questions For The Authors:**

1) How were the CoT prompts designed? Did you use task-specific examples? If so, did you add a "chain" to the CoT prompts that is similar to the decomposition process of the proposed sacrotic questioning?
2) How many self-consistency runs did you use?
3) What are the temperatures you used?

**Reasons To Accept:**

This work proposes a relatively novel inference pipeline that employs a question decomposition process and a way to add sub-question answers back to the reasoning process. The overall design is intuitive and aligns with human reasoning styles. Compared with the text-based experiments, I am more excited with the VQA experiments, because of the dynamic information query to an image, which makes VQA process explainable.

**Reasons To Reject:**

1) The evaluation datasets are not comprehensive enough. The selected domains of MMLU seem a bit arbitrary, and the relative evaluation size is small. Because few-shot inferences cannot be run multiple times, some kind of significance test would be helpful here. In addition, the MATH dataset is way too challenging for chatgpt-based models, and I am not convinced that the improvement is because the proposed sacrotic prompting actually helps it to solve certain problems, instead of merely introducing more biases to guess a small number of questions better. That being said, I think several QA datasets (e.g., StrategyQA) and simpler math datasets (e.g., GSM8K) should be evaluated because of their decomposition nature.

2) The chain of thought and self-consistency baseline is not well-described, so it is hard to compare the proposed pipeline with existing inference methods. (see questions)

3) Experiments involving more base LLMs such as the latest GPT-4 are necessary to make the overall claim, because different LLMs may adapt differently to the same inference pipeline, and this pipeline might not be able to improve over CoT when GPT-4 is used.

**Reproducibility:**

3: Could reproduce the results with some difficulty. The settings of parameters are underspecified or subjectively determined; the training/evaluation data are not widely available.

**Reviewer Confidence:**

3: Pretty sure, but there's a chance I missed something. Although I have a good feel for this area in general, I did not carefully check the paper's details, e.g., the math, experimental design, or novelty.

---

> ### Author Rebuttal · Authors · 2023-08-29
>
> We thank the reviewer for your valuable comments and suggestions. Below, we will address your questions.
>
> **Q1: Experimental results on several other QA and Math datasets:**
>
> Below we provide the performance of our method and two strong baselines on GSM8K and StrategyQA datasets. As one can observe, our method has significant performance improvement compared to baselines. We use ChatGPT with temperature 0.7 for all methods. For SC-CoT, we sample 20 reasoning paths.
> | | GSM8K | StrategyQA |
> | --- | --- | ----------- |
> | CoT | 79.0 | 59.7 |
> | SC-CoT |86.0 | 63.0 |
> | Socratic-Questioning | 89.33 | 65.33 |
>
> We tried our best to reproduce the results of CoT and SC-CoT reported in (Wang et al., 2022) on StrategyQA. Following (Wang et al., 2022), we use the question-only set from BIG-bench collaboration (2021) and use the exact same prompt template and in-context examples in SC-CoT (Wang et al., 2022). However, we cannot reproduce the results on StrategyQA in (Wang et al., 2022) since Code-davinci-002 and Code-davinci-001 are no longer publicly available. In addition, our results of ChatGPT on StrategyQA also agree with more recent studies in [1].
>
> [1] IS CHATGPT A GENERAL-PURPOSE NATURAL LANGUAGE PROCESSING TASK SOLVER? Chengwei Qin, Aston Zhang, Zhuosheng Zhang, Jiaao Chen, Michihiro Yasunaga, Diyi Yang
>
> **Q2: Design and experimental details of CoT and SC-CoT:**
>
> (1). We have 5 in-context examples for both CoT and SC-CoT on all datasets. The thought-chains are carefully designed by the authors such that it naturally follows human thinking processes which involve question decompositions, intermediate reasoning steps, and final conclusions. We will add in-context examples and prompts for CoT, SC-CoT in the appendix in the revised version. Below, we show an example of the thought-chain used in the in-context demonstrations for CoT in MMLU physics. We will add more details of CoT and SC-CoT in the revised version.
> ```
> Question:
> An object of volume 2 × 10^-3 m^3 and weight 6 N is placed into a tank of water, where it floats. What percentage of the object's volume is above the surface of the water?
> Options:
> A. 12%, B. 30%, C. 60%, D. 70%
> Answer:
> The percentage of the object's volume that is above the surface of the water can be calculated using the formula: Percentage above surface = [(Density of water - Density of object) / Density of water] × 100. In this question, the percentage above surface = [(997 kg/m^3 - 306 kg/m^3) / 997 kg/m^3] × 100 = (691 kg/m^3 / 997kg/m^3) × 100 = 0.693 × 100 ≈ 70%. Thus, the final answer is: D
> ```
>
> (2). We use 20 runs for SC-CoT. The number is based on the performance curve in (Wang et al., 2022).
>
> (3). The temperature is 0.7. We use the same temperature for all methods and all datasets.
>
> **Q3: Experiments based on recent state-of-the-art LLMs, e.g., GPT-4:**
>
> We want to first clarify that GPT-4 was not generally available when we submitted the paper, so comparison with GPT-4 was not feasible during the submission time. To showcase the generalizability of our approach, we have run CoT and Socratic Questioning on MMLU Chemistry and LogiQA based on GPT-4. The experimental results show that our Socratic Questioning approach still significantly outperforms CoT. We will add these details in the revised version.
> | | MMLU Chemistry | LogiQA |
> | --- | --- | ----------- |
> | CoT | 80.2 | 70.3 |
> | Socratic-Questioning | 85.7 | 75.3 |

---

### Official Review · Reviewer_VEuk · 2023-08-05

**Soundness:** 3

**Excitement:**

4: Strong: This paper deepens the understanding of some phenomenon or lowers the barriers to an existing research direction.

**Paper Topic And Main Contributions:**

While CoT prompting enables LLM to deal with complex problems, it mainly depends on the initial generations, which leads to the propagation of errors into the final answer. Instead, this paper presents a new method of prompting called Socratic Questioning that leverages LLM to proactively raise the sub-questions and collect information to answer the original questions in a recursive manner. The authors performed experiments on various reasoning tasks not only for language-only tasks but also for multimodal tasks.

**Questions For The Authors:**

Question A. In subsection 6.3 (lines 481-485), are there any examples of generating incorrect answers with larger depth? I think it is an interesting finding but there is less enough evidence on it.

Question B. How’s the distribution of the number of hints and depth needed to answer the question? (each for correct answer and incorrect answer)


**Reasons To Accept:**

1. The paper is well-written and the problem is well-motivated by grounding on the prior work. Comparison with other methods can help readers to understand this method more easily
2. Qualitative results have interesting findings about how the numbers of turns and depths affect the generation. This can provide insights to other future researchers who can try similar methods into other tasks that might have diverse reasoning levels
3. Results show that Socratic questioning outperforms other prior approaches in diverse tasks.


**Reasons To Reject:**

1. Compared to other approaches, this method can be inefficient in terms of generation cost although it outperformed other methods. There needs to be an experiment to compare the number of calls.
2. To support whether QA2H module is necessary, there needs to be an experiment to compare ablation study with or without this module.
3. There are unclear parts needed to generalize this prompting method into other reasoning domains. More detailed explanation on the following parts, even as additions to the appendix, would be needed. (1) In 3.2.2, there can be various sub-questions can be made to resolve the original answer but it is less concrete how the authors created simpler sub-questions in prompts. If the sub-questions are not well-designed, then it would be inefficient since there can be many useless sub-questions. (2) For the same reason, how authors designed confidence levels in few-shots is also missing.


**Reproducibility:**

3: Could reproduce the results with some difficulty. The settings of parameters are underspecified or subjectively determined; the training/evaluation data are not widely available.

**Reviewer Confidence:**

3: Pretty sure, but there's a chance I missed something. Although I have a good feel for this area in general, I did not carefully check the paper's details, e.g., the math, experimental design, or novelty.

---

> ### Author Rebuttal · Authors · 2023-08-29
>
> We thank the reviewer for your valuable comments and suggestions. Below, we will address your questions.
>
> **Q1: Experiment to compare the number of calls:**
>
> In the following table we show the theoretical maximum number of calls (row #1) in CoT, SC-CoT, ToT and Socratic Questioning in 2 and 3 turns. We also provide the empirical results of the average number of calls per instance (row #2) and average running time per instance in seconds (row #3) for all methods (we will include this table in the final submission).
>
> - For SC-CoT, we fix the number of calls to 20 times on all the datasets based on the performance curve in (Wang et al., 2022).
> - In ToT, k represents the number of thoughts allowed to be generated per step, T represents the maximum number of steps and  b represents the maximum number of states to keep at each step in BFS. Following (Wang et al., 2022), we set k=5, T=3, and b=4.
> - In Socratic Questioning, q represents the maximum number of raised sub-questions for a parent node, t represents the maximum number of turns for executing a self-questioning for a node, and d represents the largest depths for a node.
>
> As one can observe, Socratic Questioning with 2 turns and 3 turns achieves better efficiency compared to SC-CoT and ToT. The main reason is that, in the experimental dataset, most questions do not require a large amount of thinking steps to reach the correct answers. Socratic Questioning, adaptively raises sub-questions based on the complexity of the original question and arrives at the correct answer without reaching the theoretical maximum number of turns or depth. In contrast, both the SC-COT and the ToT algorithms employ fixed settings for the number of thoughts generated per step. For relatively straightforward questions, these fixed settings introduce high computational overhead, making the algorithms less efficient in these questions.
>
> |      | Standard-Prompting | CoT | SC-CoT | ToT | Socratic Questioning (2 turns) | Socratic Questioning (3 turns) |
> | ----------- | ----------- | ----------- | ----------- | ----------- | ----------- | ----------- |
> | Theoretical number of calls      |     1   |     1   |     20   | $$  k + b \times k \times (T - 1)  $$ |  $$ 3 \times \sum_{i=1}^{d-1} [q \times (t-1)]^i $$ |   $$ 3 \times \sum_{i=1}^{d-1} [q \times (t-1)]^i $$ |
> | Average calls per instance   |      1.0   |     1.0   |     20.0   | 31.1 | 9.2 |  18.7  |
> | Average running time per instance (second)   |      0.33   |     3.350   |    67.09   | 77.99 | 34.15  |  53.65 |
>
> **Q2: Ablation study for the QA2H module:**
>
> Thanks for pointing this out. We will include this ablation study in the revised version.
>
> **Q3: Regarding the design and generalization of the prompting methods to other reasoning domains:**
>
> - (1). To make our method generalize to other reasoning domains, we carefully design in-context demonstrations to guide the LLM to generate more basic sub-questions in an efficient manner. More concretely, to create high-quality sub-questions in the in-context examples, we take the human reasoning process and domain knowledge into account and carefully annotate the sub-questions by ensuring that they are more basic questions compared to the original question and their solutions can contribute to the reasoning process of the original questions.  Below, we provide a concrete example with sub-questions in the in-context demonstration in the physics domain:
> ```
> Question:
> An object of volume 2 × 10^-3 m^3 and weight 6 N is placed into a tank of water, where it floats. What percentage of the object's volume is above the surface of the water?
>
> Sub-questions:
> 1. When an object floats, what function describes the relationship between the object's volume and weight?
> 2. What is the density of water?
> 3. An object of volume 2 × 10^-3 m^3 and weight 6 N, what is the object's density?
>
> ```
> For more examples of sub-questions, please refer to Figure 12 in appendix G.2. Based on our experiments in math, physics, chemistry and VQA domains, we argue that with a few examples (5 in all our experiments) Socratic-Questioning can generalize to a new domain.
>
> - (2). Following [1], we ask the LLM itself to output a confidence level, “high”, “middle”, or “low”, towards its answer. In the in-context demonstrations, we label the correct answers with supportive hints in the context as “high” confidence, label the correct answers without supportive hints as “middle” confidence, and  label incorrect answers as “low” confidence. In this way, we can guide the model to align its confidence to the correctness of the predicted answers. Our algorithm will continue raising sub-questions if the estimated confidence is not “high”. Below is an example with low confidence. Please refer to Appendix G.1 Figure 11 for more examples.
> ```
> Question:
> How much work is required to charge a 10 μF capacitor to a potential difference of 100 V ?
> Option:
> [A. 0.005 J, B. 0.01 J, C. 0.05 J, D. 0.1 J
> Thus, the final answer is: C
> Confidence: low
> ```
>
> [1] Language Models (Mostly) Know What They Know. Saurav Kadavath∗
> , Tom Conerly, Amanda Askell, Tom Henighan, Dawn Drain, Ethan Perez,
> Nicholas Schiefer, Zac Hatfield-Dodds, Nova DasSarma, Eli Tran-Johnson, Scott Johnston,
> Sheer El-Showk, Andy Jones, Nelson Elhage, Tristan Hume, Anna Chen, Yuntao Bai,
> Sam Bowman, Stanislav Fort, Deep Ganguli, Danny Hernandez, Josh Jacobson,
> Jackson Kernion, Shauna Kravec, Liane Lovitt, Kamal Ndousse, Catherine Olsson,
> Sam Ringer, Dario Amodei, Tom Brown, Jack Clark, Nicholas Joseph,
> Ben Mann, Sam McCandlish, Chris Olah, Jared Kaplan∗
>
> **Q4: Example of generating incorrect answers with larger depth:**
>
> Our observation is that as the depth increases, the context information in the original questions start to vanish and the answers to the sub-questions may be inaccurate in the context of the original question. Thus, by adding the answers to sub-question in larger depth as hints, we can introduce noises to the reasoning process of the LLM which results in wrong answers. Below, we show a concrete example.
>
> ```
> Consider a physics question example:
> When a spinning system contracts in the absence of an external torque, its rotational speed increases, and its angular momentum [ A. decreases, B. increases, C. remains unchanged, D. may increase or decrease ]"?
> ```
>
> Socratic Questioning raises a sub-question: "What affects the rotational speed of a spinning system?" The initial answer to this sub-question is “Conservation of angular momentum”, which provides enough information to answer the original question. In a larger depth setting, the Socratic Questioning raises a deeper sub-question: “What is the relationship between rotational speed and angular momentum in a spinning system?” The answer to this question is: “The angular momentum is directly proportional to the rotational speed”. Incorporate this hint, the Socratic Questioning changes the answer of the first sub-question to: “The angular momentum is directly proportional to the rotational speed.”, which results in an incorrect final answer B.
>
> **Q5: Distribution of number of hints and depth needed to answer the questions:**
>
> Below we provide the averaged number of hints and depth used to answer the original questions for correct and incorrect answers. As one can see, for incorrect answers, the LLM raises more sub-questions for the original question which shows that LLM tends to explore more thinking space when the question is hard or it does not know the correct answer. This trend also agrees with the depth. If the question is hard for the model, the model tends to break the sub-questions into even more basic questions.
>
> |      | Description | Description |
> | ----------- | ----------- | ----------- |
> | Average Hints      | 3.28       | 3.68       |
> | Average Depth   | 2.89        | 2.92        |

---

### Official Review · Reviewer_VWnD · 2023-08-06

**Soundness:** 4

**Excitement:**

4: Strong: This paper deepens the understanding of some phenomenon or lowers the barriers to an existing research direction.

**Paper Topic And Main Contributions:**

This paper proposes SOCRATIC QUESTIONING, a divide-and-conquer style algorithm that mimics the recursive thinking process to solve complex reasoning tasks. The algorithm consists of (1) an top-down exploration process that decomposes the question into simpler sub-questions until the sub-questions are answered confidently, and (2) a bottom-up backtracking process that collects the answers of sub-questions and solve the high-level tasks. All the modules are implemented by prompting large language models without additional model training. The proposed algorithm is evaluated on both language-only and multimodal tasks. The results show that SOCRATIC QUESTIONING is able to achieve better accuracy on solving complex reasoning tasks compared to several SOTA approaches. The paper is clearly written and easy to follow.

**Questions For The Authors:**

Question 1: I notice in Table 1, the results of ToT are quite low, and I guess a possible reason is that you apply "exact matching" to extract the answers. As shown in Figure 4, the results of ToT might be correct but the format is different. Have you tried to loose the condition of such "exact matching" and compute the accuracy?

Question 2: It seems in some cases introducing more turns does not bring improvements, and even might hurt the performance. This is quite surprising to me since intuitively more turns of reasoning at least should not reduce the accuracy. Do you have any analysis on this?


**Reasons To Accept:**

- Interesting idea of mimicking recursive thinking process to solve complex reasoning tasks by leveraging LLMs;
- Both language-only and multimodal tasks are included, showing it's ability on solving a wide range of tasks;
- Paper is well written and easy to follow. The analysis is relatively thorough.

**Reasons To Reject:**

- As recursive prompting introduces additional computational complexity, I would like to see some analysis on the algorithm efficiency;
- The language-only tasks in the current paper mostly leverage MMLU tasks, and I think including some complicated real-world tasks (e.g., Game of 24, puzzle) would make the work more interesting.

**Reproducibility:**

3: Could reproduce the results with some difficulty. The settings of parameters are underspecified or subjectively determined; the training/evaluation data are not widely available.

**Reviewer Confidence:**

3: Pretty sure, but there's a chance I missed something. Although I have a good feel for this area in general, I did not carefully check the paper's details, e.g., the math, experimental design, or novelty.

**Typos Grammar Style And Presentation Improvements:**

I would suggest to add a concrete example in Figure 3 to make the understanding of the algorithm easier.

---

> ### Author Rebuttal · Authors · 2023-08-29
>
> We thank the reviewer for your valuable comments and suggestions. Below, we will address your questions.
>
> **Q1: Evaluation of computational cost:**
>
> In the following table we show the theoretical maximum number of calls (row #1) in CoT, SC-CoT, ToT and Socratic Questioning in 2 and 3 turns. We also provide the empirical results of the average number of calls per instance (row #2) and average running time per instance in seconds (row #3) for all methods (we will include this table in the final submission).
>
> - For SC-CoT, we fix the number of calls to 20 times on all the datasets based on the performance curve in (Wang et al., 2022).
> - In ToT, k represents the number of thoughts allowed to be generated per step, T represents the maximum number of steps and  b represents the maximum number of states to keep at each step in BFS. Following (Wang et al., 2022), we set k=5, T=3, and b=4.
> - In Socratic Questioning, q represents the maximum number of raised sub-questions for a parent node, t represents the maximum number of turns for executing a self-questioning for a node, and d represents the largest depths for a node.
>
> As one can observe, Socratic Questioning with 2 turns and 3 turns achieves better efficiency compared to SC-CoT and ToT. The main reason is that, in the experimental dataset, most questions do not require a large amount of thinking steps to reach the correct answers. Socratic Questioning, adaptively raises sub-questions based on the complexity of the original question and arrives at the correct answer without reaching the theoretical maximum number of turns or depth. In contrast, both the SC-COT and the ToT algorithms employ fixed settings for the number of thoughts generated per step. For relatively straightforward questions, these fixed settings introduce high computational overhead, making the algorithms less efficient in these questions.
>
> |      | Standard-Prompting | CoT | SC-CoT | ToT | Socratic Questioning (2 turns) | Socratic Questioning (3 turns) |
> | ----------- | ----------- | ----------- | ----------- | ----------- | ----------- | ----------- |
> | Theoretical number of calls      |     1   |     1   |     20   | $$  k + b \times k \times (T - 1)  $$ |  $$ 3 \times \sum_{i=1}^{d-1} [q \times (t-1)]^i $$ |   $$ 3 \times \sum_{i=1}^{d-1} [q \times (t-1)]^i $$ |
> | Average calls per instance   |      1.0   |     1.0   |     20.0   | 31.1 | 9.2 |  18.7  |
> | Average running time per instance (second)   |      0.33   |     3.350   |    67.09   | 77.99 | 34.15  |  53.65 |
>
> **Q2: Adding more complicated real-world tasks (e.g., Game of 24, puzzle):**
>
> Thanks for the suggestions. Although we already have a wide spectrum of tasks to support the effectiveness of our proposed method, we agree that adding more complicated real-world tasks will make this work more interesting and we will include those tasks in the revised version.
>
> **Q3: Evaluation when the answers have different formats as the gold answer:**
>
> We want to first clarify that all questions in MMLU Physics, MMLU Chemistry, and LogiQA are multiple-choice questions and the answer is always a single letter like “A”, “B” or “C”. To easily parse the model’s final output, we use “Thus, the final answer is:” as the prefix for the final answers (A or B or C or D, ect.) in the in-context examples for all methods. When we parse the output, we first run a template-based method to extract the answers after “Thus, the final answer is:”. For a few instances (12.52% in CoT, 16.4% in ToT and 11.64% in Socratic Questioning on average) that do not match the template as shown in Figure 4 ToT, the authors manually compare the model’s predictions to the ground truth answers. Thus, we assure that the final performance of all methods is not affected by the output formats.
>
> **Q4: Why the performance of ToT is quite low:**
>
> - In ToT, the number of thoughts is a fixed hyperparameter, rather than a value decided by the LLM for different questions. This static configuration is suitable for problems like the "Game of 24," where each thought leads to a fixed number of sets of subsequent states. However, for real-world problems, fixing the number of thoughts at every step may force the model to raise some subsequent thoughts that are less relevant and introduce noisy thought chains.
> - Moreover, the state evaluator in ToT fails to identify these noisy thought chains and consequently returns inaccurate scores. As a result, the noisy thought chains are kept and affect the later thought generation.
> - Furthermore, ToT accumulates errors from initial thoughts, as subsequent thoughts build upon previous inaccurate thoughts without additional guidance to correct them. This inherently flawed structure undermines the reliability of the later thought paths.
>
> We use the following example to illustrate the problems of ToT. Considering a simple physics question:
> ```
> In the absence of air resistance a ball of mass m is tossed upward to reach a height of 20 m. At the 10-m position half way up the net force on the ball is [ A. 2mg, B. mg, C. mg/2, D. mg/4 ] ?
> ```
> The correct answer is directly obtained by multiplying the mass by the gravity acceleration. ToT initially identifies the correct approach, however, due to the requirement of generating a fixed number of thoughts, it subsequently generates an additional incorrect thought: "To find the net force on the ball at the 10-m position, we can use the equation for net force. Since the ball is halfway up, the net force will be equal to half of the weight of the ball. Therefore, the net force on the ball at the 10-m position is mg/2." This incorrect thought receives a slightly higher evaluation score (0.9) from the “State Evaluator” component of ToT than the correct answer (0.8). As a result, ToT accumulates this incorrect information into subsequent calculations, leading to an ultimately incorrect conclusion.
>
> **Q5: The effect of the number of turns to performance:**
>
> Due to the calibration error in LLMs [1], sometimes the pre-trained model’s confidence is not aligned with the answer’s correctness. Thus, in such cases, the model predicts “low” or “medium” confidence in correct answers in the early turns and hence misses the correct answers. If we use fewer turns, we can keep the answer in the early turn regardless of the confidence and hence alleviate the calibration error. Below we show a concrete example in which the model predicts the correct answer in 2 turns and predicts the incorrect answer in 3 turns. When we increase the number of turns, Socratic Questioning may raise some less relevant sub-questions and hence introduce noisy information in the reasoning process. This noisy information can confuse the model, leading to incorrect responses to the original question.
>
> For example, consider a simple physics question:
> ```
> The speed of sound is slightly greater on a [ "A. cold day", "B. hot day", "C. day with steady temperature", "D. None of these"]?
> ```
> - In a 2-turn setting, our approach obtains hints: (1) "The speed of sound increases with increasing temperature.", and (2) "Humidity is a factor in the speed of sound." According to the hints, it is obvious that the correct answer is B, which is chosen by our approach in the second turn with the "middle" confidence.
> - In a 3-turn setting, since the LLM does not assign “high” confidence to the answer in the 2 turn, our approach goes deeper in the third turn and gets more information (e.g., (3) "The speed of sound can be affected by several factors, including temperature, humidity and density of the medium.", (4) "The speed of sound depends on the density and elasticity of the medium it is traveling through, in terms of physical properties.", (5) "The speed of sound increases with humidity as a result of increased air density.") As a result, by considering more hints, we potentially introduce less relevant information to the LLM and the noisy information causes the LLM to change its answer to D.
>
> **Q6: Adding concrete example to Figure 3:**
>
> Thanks for the suggestion! We will make sure to add a concrete example in Figure 3 to make the algorithm more understandable.
>
> [1] How Can We Know When Language Models Know? On the Calibration of Language Models for Question Answering. Zhengbao Jiang, Jun Araki, Haibo Ding, Graham Neubig

---

### Meta-Review · Area_Chair_PA3r · 2023-09-16

**Recommendation:** 5

**Metareview:**

This paper proposes SOCRATIC QUESTIONING, a divide-and-conquer style algorithm that mimics the recursive thinking process to solve complex reasoning tasks. The algorithm consists of an exploration process that decomposes the question into simpler sub-questions until the sub-questions are answered confidently. All the modules are implemented by prompting large language models. The proposed algorithm is evaluated on both language-only and multimodal tasks.


Based on my reading of the reviews, the reviewers like the idea, the findings, and the writing of the work.


Here are the important points raised in the reviews:
(1) computational complexity of the proposed approach, which is addressed.
(2) not evaluated on a diverse range of tasks (e.g., the game of 24 and StrategyQA): the author has agreed but has promised to address
(3) Ablation on QA2H module: promised to include, though little shown in the rebuttal.


The ratings are high for this, and I support its acceptance.
I worry that the authors have promised many changes to be incorporated in their revision (experiment on complexity, ablation  QA2H+hints, additional datasets, etc.), and I hope they keep their promise.

---

### Decision · Program_Chairs · 2023-10-07

**Decision:**

Accept-Main

**Comment:**

This paper proposes SOCRATIC QUESTIONING, a divide-and-conquer style algorithm that mimics the recursive thinking process to solve complex reasoning tasks. The algorithm consists of an exploration process that decomposes the question into simpler sub-questions until the sub-questions are answered confidently. All the modules are implemented by prompting large language models. The proposed algorithm is evaluated on both language-only and multimodal tasks.


Based on my reading of the reviews, the reviewers like the idea, the findings, and the writing of the work.


Here are the important points raised in the reviews:
(1) computational complexity of the proposed approach, which is addressed.
(2) not evaluated on a diverse range of tasks (e.g., the game of 24 and StrategyQA): the author has agreed but has promised to address
(3) Ablation on QA2H module: promised to include, though little shown in the rebuttal.


The ratings are high for this, and I support its acceptance.
I worry that the authors have promised many changes to be incorporated in their revision (experiment on complexity, ablation  QA2H+hints, additional datasets, etc.), and I hope they keep their promise.